# Does the End Justify the Means? The Role of Organizational Communication among Work-from-Home Employees during the COVID-19 Pandemic

**DOI:** 10.3390/ijerph18083933

**Published:** 2021-04-08

**Authors:** Margherita Zito, Emanuela Ingusci, Claudio G. Cortese, Maria Luisa Giancaspro, Amelia Manuti, Monica Molino, Fulvio Signore, Vincenzo Russo

**Affiliations:** 1Department of Business, Law, Economics and Consumer Behaviour “Carlo A. Ricciardi”, Università IULM, Via Carlo Bo 1, 20143 Milan, Italy; margherita.zito@iulm.it (M.Z.); vincenzo.russo@iulm.it (V.R.); 2History, Society and Human Studies Department, University of Salento, Via di Valesio 24, 73100 Lecce, Italy; fulvio.signore@unisalento.it; 3Department of Psychology, University of Turin, Via Verdi 10, 10124 Turin, Italy; claudio.cortese@unito.it (C.G.C.); monica.molino@unito.it (M.M.); 4Department of Education, Psychology, Communication, University of Bari, Palazzo Chiaia Napolitano, Via Crisanzio 42, 70121 Bari, Italy; maria.giancaspro@uniba.it (M.L.G.); amelia.manuti@uniba.it (A.M.)

**Keywords:** COVID-19, remote working, organizational communication, technostress, self-efficacy, psycho-physical disorders

## Abstract

During the first months of 2020, the world, and Italy at an early stage, went through the COVID-19 emergency that had a great impact on individual and collective health, but also on working processes. The mandatory remote working and the constant use of technology for employees raised different implications related to technostress and psycho-physical disorders. This study aimed to detect, in such a period of crisis and changes, the role of organizational communication considering the mediating role of both technostress and self-efficacy, with psycho-physical disorders as outcome. The research involved 530 workers working from home. A Structural Equations Model was estimated, revealing that organizational communication is positively associated with self-efficacy and negatively with technostress and psycho-physical disorders. As mediators, technostress is positively associated with psycho-physical disorders, whereas self-efficacy is negatively associated. As regards mediated effects, results showed negative associations between organizational communication and psycho-physical disorders through both technostress and self-efficacy. This study highlighted the potential protective role of organizational communication that could buffer the effect of technostress and enhance a personal resource, self-efficacy, which is functional to the reduction of psycho-physical disorders. This study contributed to literature underlying the role of communication in the current crisis and consequent reorganization of the working processes.

## 1. Introduction

### 1.1. Communication during the Health Emergency of COVID-19

During the first months of 2020, the world, and in particular Italy at an early stage, went through an event, the COVID-19 health emergency, that has changed the lives of all of humanity. The COVID-19 pandemic has had a great impact on health, economy, and society, causing difficulties and uncertainty for many people in several areas. The security measures adopted by the government, aimed at protecting, as far as possible, the health of everyone, mostly involved social distancing, which is considered the most effective way to manage the spread of the virus. From an economic point of view, to avoid the blocking of the productivity of both private companies and public administration, workers have turned to teleworking and remote working, in some cases even without a proper training, leading to some negative implications, in terms of work load management and technostress [1]. In addition to the use of agile working practices, the emergency measures that have been adopted are the Extraordinary Wages Fund, forced holidays, and, in some cases, even business closures [2]. 

The national economic situation has had a great influence on the working conditions, on job modalities and practices, and therefore on employees’ health and wellbeing. In this crisis situation, human resources managers could have a main role in supporting the organization and employees by conveying motivating messages, keeping workers confident toward the future and, therefore, productive [3]. 

Accordingly, the way in which the organization communicates with and supports employees in approaching change, has a direct impact on the attitudes and strategies they use to address this process. When workers recognize that a change could be positive for the organization, they are inclined to support it with committed behaviors and positive energy [4,5,6]. Especially during times like the current one, characterized by the pandemic and by its direct consequences, careful attention to the development and management of human resource practices is vital to convey support, encouragement, and job security to all employees [3].

Drawing from such considerations, the aim of this study was to consider the role of organizational communication in the present crisis and change the situation, specifically considering the mediating of technostress (as a source of stress associated to the use of technology), and of self-efficacy, with the presence of psycho-physical disorders as outcome.

This study was addressed to contribute to the reading of such a difficult period, in which communication appears to be highly relevant in organizations to manage the stress that might derive from the use of technology, that is also responsible for other new forms of negative psychological and physical consequences. In view of the above, this study attempted to extend the knowledge on this new form of stress, namely technostress, and on the important role played by organizational communication in buffering the stressing and challenging demands brought about by the current situation of change in the workplace.

### 1.2. Traditional Organizational Communication and Digital Communication 

Within the complex framework of change depicted above, organizational communication is crucial. Adopting the perspective of the ‘Organization as Communication’, the organization is considered as a complex network of communication habits, where the organization is the factual structure and the collective communicative behaviors of the employees make the organization [7]. Employees exchange information about their organization, about work and the achievement of specific and wider goals [8,9]. Messages shared within the organization can be both vertical and horizontal, namely messages exchanged between people who hold a different position within the organization or between people who occupy the same position. In both cases, communication can be formal or informal [9,10]. To have an open line of communication involves a flow of information about employees’ beliefs and thoughts [11]. 

Scientific contributions in the field describe organizational communication according to its functions [10,12]; in particular, De Nobile et al. [13] distinguished four functions: directive, supportive, cultural, and democratic communication. Directive communication refers to messages aimed at persuading, influencing, and generally at controlling employees; these concepts are in accordance with the studies by different scholars in which they describe the control function and the maintenance function, respectively [8,12,14]. Supportive communication is aimed at conveying messages useful to encourage and reassure individuals [10,15,16]. Cultural communication is intended to share the internal rules of the organization with the employees, its purpose is to join in and inform newbies [17,18]. Finally, democratic communication refers to the involvement of members of the organization in decision-making processes. 

Beyond the most traditional definitions, organizational communication can be considered a key important element to involve employees and to enhance their commitment, with positive outcomes for both the organization and the individual [19]. Considering its main functions, beside being addressed to manage the information flow, organizational communication aims to motivate and support human resources especially during periods of change and crisis like the one considered in this study. 

Drawing from this perspective, this study argued for the role of organizational communication in the pandemic; in the current scenario, in fact, forced distancing could decrease individual well-being levels and deplete relationships. 

Yet, during organizational restructuring, communication strongly influences employees’ commitment [19,20], their trust toward the organization [21,22] and their attitude towards change [23]. Accordingly, Rogiest et al. [20] showed that the quality of communication might convey emotional commitment to change. Good informal communication might reduce employees’ sense of uncertainty, fostering involvement and participation. A participatory and more informal communication modality could be effective in reducing uncertainty when workers are called to develop new skills [24,25,26] and when change requires the construction of new roles.

Therefore, a well-designed plan for organizational communication, namely allowing employees to be informed and to feel involved about the future of the organization, could be functional to reduce discomfort, uncertainty, and consequent negative emotions. In this vein, being a central aspect of management [27], communication could be strategically adopted by organizations both to manage and convey information related to job activities and requirements but also to strengthen employees’ sense of belonging and identification. Thus, empirical evidences confirmed that employees’ perceiving that the organization efficiently communicate with them would tend to report higher levels of job satisfaction [28], better performance [29], and lower levels of stress [30].

**Hypothesis** **1.**
*Organizational communication is negatively associated with psycho-physical disorders.*


The massive growth of technological innovations (information technology, telecommunications, consumer electronics) that has infested our lives in the past few decades has contributed to shape what has been called an information revolution [31]. Certainly, this evidence has impacted also on the management of organizational communication. The applications of information technology, communication, and the digitalization of work have become valuable tools used in many work sectors. Information and Communications Technology (ICT) allows an improvement of work processes and facilitates access to knowledge, in an era where mediated communication from computers (CMC) are rapidly becoming part of our everyday life.

In line with this perspective, ICT has been described as organizations providing complete and quick information, and blurring hierarchical levels and the borders of departments [32], as for instance through a pervasive use of synchronous and asynchronous messaging, that allow people to cooperate and coordinate almost in real time [33]. Scholars in the field showed controversial results regarding the change in the means of communication in which, if on the one hand face-to-face communication is considered as the ideal, on the other it may not be in certain situations [34]. Undoubtedly, ICT have shown positive implications for the people who use them, such as the breadth of information available, the rate of processing data and the reliability of data [35].

### 1.3. The Role of Technostress and of Self-Efficacy on Psycho-Physical Disorders

An evident and fundamental advantage brought about by ICT is that they have allowed new forms of work, such as teleworking. Unfortunately, there are also negative implications in the use of these technologies: negative feelings are mainly linked to the presence of high levels of stress in workers caused by the feeling of being connected and performative 24 h a day [36]. In this framework, performance of difficult tasks can drop and stress and workaholism can take over [37]. 

A recent definition of technostress highlighted that it can be considered as “the stress that users experience as a result of application multitasking, constant connectivity, information overload, frequent system upgrades and consequent uncertainty, continual relearning and consequent job-related insecurities, and technical problems associated with the organizational use of ICT” [37]. The phenomenology of technostress is to be identified with the symptoms of anxiety, physical disorders, such as mental weakness, poor concentration, feeling of tiredness and inability to sleep [38,39]. Technostress traces back to the use of technology in job demands, above all, information overload is a significant source of stress, due to the high information load to manage that can affect the lack of control of such a large number of stimuli [40]; moreover, the feeling of having to be always and everywhere connected and ready for a response thanks to the use of ICT has an important role in the causes of stress [41]. Other stress-inducing factors are due to the intensity of teleworking and frequent breaks while doing it, to the large amount (but low quality) of e-mail communication [42,43,44]. In the literature, the classification of stressors is the one carried out by Tarafdar and collaborators [37] who identified five dimensions that contributed to explain the factors leading to technology-related stress: they are techno-overload [45], potentially linked to quicker and longer work overload than usual; techno-invasion [46], relating to blurred work-life boundaries; techno-complexity [47] referred to the difficulty in using ICT that makes the worker feel a disparity between the use of these and his or her skills; techno-insecurity [48], describing situations in which a person is afraid to lose his or her job as a consequence of the perceived inability to master ICT or to be replaced by them; and the techno-uncertainty [49] related to the dynamism of ICT which are constantly updated forcing the worker to constantly learn how they evolve. In the study by Molino et al. [35], the authors proposed a short version of this tool considering three dimensions, namely techno-overload, techno-invasion, and techno-complexity, suggested by the authors as the most relevant in this period of emergency remote work. In the particular scenario characterized by the pandemic, the psycho-physical well-being of workers has assumed a key role: aspects such as the overload due to the use of technology, the more blurred boundaries between work and private life due to teleworking but also the difficulty in learning how to use new working tools, can be stress creators for employees [35]. 

Considering this scenario, specific attention should be paid to the consequences of technostress on health. A study by Reinecke et al. [50] suggested that the perceived stress due to the digitalization of communication process, is responsible for burnout and psychological disorders such as anxiety and stress. Moreover, as underlined by previous studies on this topic, technostress was found to be responsible for several psychological and physical disorders. As psychological disorders, they are depicted to include symptoms such as anxiety, technophobia, panic, mental fatigue, frustration, depression, and sleeping trouble [39,51,52]. As for physical disorders, several studies described symptoms related to headaches, muscle cramp, stomach and intestinal problems, heart attack, high blood pressure, and insomnia [52,53]. Moreover, other empirical evidences assessed the physiological aspect related to the use of ICT. A study by Riedl [54] suggested that the interaction between ICT and individuals could result in increased levels of adrenaline and cortisol, labeled as stress hormones, and in an increased activity of the cardiovascular system. These results showed evident consequences on individuals’ health. Similar contributions investigated the effects of technostress through the measurement of the alpha-amylase hormone, suggesting a variance on the performance and on self-reported data on stress [55]. Moreover, an interesting study by Gallugh et al. [56] considered the negative consequence of technostress, by measuring the alpha-amylase hormone as an objective indicator of strain. Results reported that ICT were responsible for stress and strain. However, findings suggested that the variable labeled as resource control, considered a coping strategy, was shown to moderate strain. Therefore, the study suggested that resources might play a crucial role supporting individuals in managing work stress. 

According to the authors of the Job Demands–Resources model [57], two main factors interact in the working context: job demands described as “stressful, physical, psychological, social, or organizational aspects of a job that require effort and can cause an energy exhaustion”; and job resources considered as the “physical, psychological, or social aspects of a job that stimulate growth and help people to achieve their goals” [57] (p. 2). Considered individually, these aspects have opposite consequences: on the one hand, job demands can lead to burnout outcomes as their overload might cause a worsening of the worker’s health; on the other hand, job resources represent a catalyst for motivation processes and can balance job demands, for this reason they are functional to the achievement of specific objectives [2]. In relation to the intensity of the outcomes, the consequences of the use of technology in the working contexts can be considered both demands and resources. 

In light of the Job Demands-Resources Model [57], beyond job resources, individuals might also rely upon a number of personal resources, self-efficacy being one of the main one. Yet, personal resources are described as positive aspects of the self, linked to resilience and to the ability of individuals to control and manage their environment [58]. These resources have a positive impact on psychological and physical well-being, supporting individuals in dealing with demanding situations, keeping them energetic and protecting them from psychological discomfort [59,60].

Self-efficacy plays an important role in stress and work studies. The main assumption is that exposure to stressful factors has no negative consequences, if the person maintains higher levels of control; however, if exposure to stressors occurs when the person is not able to control it, exposure to stressors could have damaging consequences [61]. According to Bandura’s Cognitive Social Theory, low levels of self-efficacy in controlling certain situations are associated to the stressful experience. Depression, anxiety, helplessness, and pessimistic thoughts about one’s own performance and that of others, are all feelings related to the presence of low levels of self-efficacy [62]. According to the demand-control model [63], tiredness, depression, and physical illnesses are due to a low control of the situation by the worker and a high level of environmental demands. Furthermore, it is important to specify that not all mental and physical disorders are specifically related to the job role, there are several factors influencing them [64,65]. According to the transaction-based model [66] and the person-environment fit model [67,68], when individual resources are not sufficient to satisfy job demands, the psychological and behavioral response is stress.

As a personal resource linked to resiliency, self-efficacy can be the starting point for dealing with stressful experience and for improving the work environment and the work participation [69]. The scientific literature, in particular studies by Bandura [70,71,72], suggested that self-efficacy concerns the degree of control that an individual has over himself or herself and over the situation in which he or she is involved: the person’s beliefs about this aspect will lead him or her to choose the goals to be pursued and the level of effort and commitment to be spent. Self-efficacy has an influence on how the social work context [73] is perceived: it refers to the fact that each role within the organization raises expectations about how the person who holds that particular role should behave; therefore, the perception of the social working context reflects the individual perception on the behavior of the people working in the organization (supervisor, top management, and colleagues). The members of the organization are the social “frame of reference” [74] to which every employee refers and allow them to receive and exchange information, to make sense of their work and therefore to actively participate in work contexts in a satisfactory dynamic, traditionally linked to well-being [75].

In line with this evidence, the following hypotheses were formulated:

**Hypothesis** **2a.**
*Technostress is positively associated with psycho-physical disorders.*


**Hypothesis** **2b.**
*Self-efficacy is negatively associated with psycho-physical disorders.*


### 1.4. The Impact of Organizational Communication on the Employees’ Self-Efficacy and Well-Being

Organizational communication is a crucial aspect of working life. It is important for organizations to manage processes, to control procedures, to inform workers on job requirements, tasks and roles, and for workers to socialize with the context, to feel part of the organization, to collaborate with others, and to learn and transfer knowledge and skills. Abundant research confirmed these evidences stressing the need to adopt a communication style based on active listening, on participation to develop team effectiveness and to promote teamwork [76]. To be effective, communication should be carefully planned and managed [77,78]: keeping open communication with managers could assure effectiveness as long as it could improve performance, strengthen identification [79,80,81], support stress management [82,83], and coping with change [84,85]. Empirical evidence supported a positive correlation between communication and employee’s performance: whereas communication is characterized by openness, performance feedback, and information about procedures [78,85,86,87,88]. At the same time, few studies have specifically addressed yet the communicative relationship between management and employees. To this purpose, perceived organizational support could be a meaningful variable in this process. 

Perceived organizational support (POS) [89,90,91,92] refers to the perception that workers have about how they are ensured and supported in the company they belong to. Allen’s studies [93,94] show a strong relationship between the perceived support and the communication style adopted by managers. Organizational support theory [90,91,92] explains that workers tend to create a general positive evaluation of their performance in order to meet their socio-emotional needs and in order to settle if the organization reward their work achievements and help them in times of need [95]. An interesting point of view concerns the fact that when managers have an open line communication and there is a good level of perceived support by employees, they tend to adopt positive behaviors that could be beneficial to the organization as they feel obliged to reciprocate that behavior [95]. The context of this relationship could be the organizational community. Bauman [96] described communities as those places where one can positively confront each other on different topics, places where mutual help is not considered a duty. The members of the in-group identify with each other, create an image of their new self within the community by developing a process of self-identification with the organization [97] that leads to positive outcomes for the company, such as, for example, the active participation in the community [98]. The identification process guides the development of greater commitment to the organization: in particular, this happens when the individual feels that the organization reflects his or her identity, positively affects his or her self-confidence and social position [99]. 

Moreover, organizational communication is described to have an important role in understanding specific situation that workers are experiencing [100]. This would involve an improved sense of control over the situation, the meaning of what workers are doing, and, more importantly in the awareness process, it can be helpful in the identification and understanding of the source of stress [100,101].

Furthermore, communication in organization is recognized to be helpful in supporting employees in efficiently coping with work stressors [82] and with organizational change [83]. In line with previous studies, communication is related with well-being outcomes, it is also recognized to be a potential buffer for technostress, since supporting workers with information helps reducing the anxiety and discomfort associated with technology [102].

Another important and positive element on communication that has be underlined, is that communication can enhance emotional commitment and participation, which, in turn, can decrease uncertainty and encourage individuals to develop skills [24,25,26]. Communication, in fact, is a powerful source of interconnection ensuring personal growth and organizational performance [103]. Communication is depicted to have a key role for the team success, helping people in sharing meanings, and in collaborating for effective progress [104,105]. Moreover, according to the Job-Demands Resources model, communication, among other job resources, such as support from the environment and autonomy, was found to be linked to occupational safety, to engagement, to commitment, and to satisfaction [106,107]. In this light, organizational communication can be considered a job resource [95,108] because several evidences confirm that whenever perceived to be effective and supportive, employees enhance engagement, positive emotions, well-being, their sense of autonomy, competence and performance tend to increase [57,95,109]. Therefore, a combination between organizational and personal resources could contribute to reinforce a positive relationship leading to a continuous growing and development [110].

Considering the literature and the relationships between communication and the several individual and organizational processes described above, the following hypotheses were formulated:

**Hypothesis** **3a.**
*Organizational communication is negatively associated with technostress.*


**Hypothesis** **3b.**
*Organizational communication is positively associated with self-efficacy.*


### 1.5. Psycho-Physical Disorders at Work and during Pandemic of COVID-19 

The lockdown, and the consequent restrictions, have had a great impact on the health of citizens; in particular, with regard to workers, the consequences relating to social distancing and working from home have affected the development of psycho-physical symptoms, especially for those people who were alone and already psychologically off [111]. In a study by Cuiyan et al. [112], conducted in China already in the initial phase of the lockdown, the psychological impact was assessed as moderate to severe, and about one third of the population reported feeling anxious. Several studies have shown that psychological pains can be the cause of structural and functional changes in the hippocampus and of hormone levels change in the human body [113,114]; moreover, psychological stress is depicted to have an impact on high blood pressure and hypertension levels [115,116]. From the research by Janula et al. [111] on the consequences of the pandemic on healthy workers, it was found that many participants reported having headaches, indigestion, and sleep irregularities during the period of lockdown. Other research conducted in London also showed that nearly two thirds of respondents said they did not have good sleep quality since the start of closure [117]. Furthermore, both gastrointestinal problems and palpitations have a strong correlation with mental health: negative emotions can lead to the development of intestinal tract disorders [118,119,120] and even increased heart rate [121]. 

In the light with prior contributions and with the most recent evidences emerging from research on individuals and on workers discomfort and wellbeing related to the COVID-19 pandemic emergency, this study aimed to understand, in a protective perspective, the role of organizational communication and its relation to the more and more pervasive presence of technology in daily life and the consequences on health. 

For this reason, this study hypothesized also indirect effects from organizational communication:

**Hypothesis** **4.**
*Organizational communication is indirectly and negatively associated with psycho-physical disorders (4a) through technostress; (4b) through self-efficacy.*


Figure 1 shows the theoretical model and the expected relationships and hypotheses.

## 2. Materials and Methods 

### 2.1. Sample and Procedure

The sample was composed by 530 Italian workers (60.4% of them were females and 39.6% were males). Participants reported an average age of 44 years (SD = 8.70), most of them were married or cohabiting (76.2%), and had a university education or higher education (63.7%). Moreover, participants had mainly a permanent contract (73.1%), fewer had fixed-term contracts (26.9%), they were mainly employees (83.4%), followed by managers and executive managers (16.8%), and they belonged to both private (59.6%) and public (40.4%) organizations. 

Participants in the study completed the questionnaire during the lockdown imposed by the Italian Government for the COVID-19 pandemic emergency (from 9 March 2020 to 3 May 2020). All participants reported working from home at that time. To control and to be sure that participants were working from home, a specific question was asked about their working situation. All the participants who reported that they were not working from home were not considered in this study, and all the participants who reported that they were working from home were maintained.

Data were collected through an online self-report questionnaire, containing a cover letter explaining: how to complete the form, the voluntary participation in the study, and the anonymity, together with an explanation of data processing and privacy according to the Italian code of ethics of the order of psychologists. All participants provided their informed consent in a specific box before filling in the questionnaire. The research observed the Helsinki Declaration (World Medical Association [122]) and the General Data Protection Regulation. The ethical approval was not necessary because the study did not provide medical treatments or other practices that can be cause of psychological or social malaises to participants.

### 2.2. Measures

The questionnaire encompassed the following measures related to the variables considered in the study. 

*Organizational communication* was measured using 3 items taken from the Copenhagen Psychosocial Questionnaire (COPSOQ) developed by Kristensen and Borg [123]. An example item is: “It’s easy to get the information you need”. The reliability coefficient (α) in this study is 0.73. Participants were asked to indicate the occurrence of each attitude or behavior described by the items using a 5-point Likert scale from 1 (never) to 5 (always).

*Technostress* was measured using the Italian version of the scale developed by Ragu-Nathan et al. [1], and adapted by Molino et al. [35]. In this case, participants were invited to express their agreement or disagreement with each item using a 5-point Likert scale from 1 (strongly disagree) to 5 (strongly agree). The Italian brief version of the scale has 11 items and considers three dimensions of technostress: techno-overload (four items; e.g., “I am forced by technology to do more work than I can handle”; α = 0.89), techno-invasion (three items; e.g., “I have to be in touch with my work even during my vacation due to technology”; α = 0.81), and techno-complexity (four items, e.g., “I do not know enough about technology to handle my job satisfactorily”; α = 0.90). The overall 11-item scale has α = 0.90.

*Self-efficacy* was measured using 3 items from the Italian Psychological Capital Questionnaire developed by Alessandri et al. [124]. An example item is: “When I analyze a problem, I am confident that I will find a solution”. The reliability coefficient (α) in this study is 0.76. The Likert scale adopted to measure the agreement or disagreement of participants with each item was a 6-point one, ranging from 1 (strongly disagree) to 6 (strongly agree).

*Psycho-physical disorders* were measured using 11 items from the scale developed by Ilmarinen [125]. Respondents were asked to think about their experience from the beginning of home isolation during the lockdown period, and to indicate the occurrence of each of items related to their specific psycho-physical situation using a 6-point Likert scale from 1 (never) to 6 (always). An example item is: “Psychological and mood disorders (e.g., Depression, anxiety, panic attacks, obsessions, etc.)” or “Gastro-intestinal disorders (e.g., gastritis, pancreatitis, irritable colon, intolerances, etc.)”. The reliability coefficient (α) in this study is 0.84.

All measures used Likert scales. The items extracted from the COPSOQ measure, originally conceived to assess the occurrence of behaviors, were also adapted to a Likert scale, as previously done by other studies in the Italian context [126,127,128]. Overall, according to Becker and Ismail [129], we used different Likert scales within the same model. Moreover, beyond reliability, also discriminant validity was confirmed through the Fornell–Larcker criterion [130] and the Heterotrait–monotrait HTMT [131] ratio of correlation performed. Results of the discriminant validity are reported in Section 3.

### 2.3. Data Analyses

Data analyses related to correlations (Pearson’s r), alpha reliabilities (α) for each scale, and also descriptive statistics were performed with SPSS 27. A Structural Equations Model (SEM) was estimated with MPLUS 8, in order to test the mediating role of technostress and of self-efficacy between organizational communication and psycho-physical disorders. Hypotheses were specified a-priori and a partial mediation model was performed [132]. Goodness of fit of the model was evaluated by the chi-square value (χ^2^), the Comparative Fit Index (CFI), the Tucker Lewis Index (TLI), the Root Mean Square Error of Approximation (RMSEA), and the Standardized Root Mean Square Residual (SRMR). Indirect effects were also assessed through a bootstrapping procedure which extracted 2.000 new samples to calculate direct and indirect parameters of the model [133]. 

Considering the high number of items, the latent variables of technostress and of psycho-physical disorders were built with the parceling method and, respectively, latent variables were composed by three and two parcels (indicators composed by two or more items on average). The parceling method can reduce type I errors in item correlations and can reduce the likelihood of a-priori model misspecification [134,135]. All parcels showed significant loadings (*p* < 0.001) in the present SEM. 

Moreover, to examine the potential effects of common method bias, two different models were compared following Harman’s single-factor procedure [136]. First, a confirmatory factor analysis considering the four latent variables was conducted, obtaining the following fit indices: χ^2^(137) = 337.428, *p* < 0.001, CFI = 0.96, TLI = 0.95, RMSEA = 0.05, SRMR = 0.07; then it was compared with a one-factor model with all items loading on one factor, which obtained the following fit indices: χ^2^(151) = 2637.081, *p* < 0.001, CFI = 0.51, TLI = 0.45, RMSEA = 0.176, SRMR = 0.132, showing that the first model fitted the data better than the one-factor model, thus supporting the appropriateness of each item related to the hypothesized latent factor. Additionally, a chi-square statistical significance comparison confirmed this result (chi-square difference = 2378.173 with 5 df; *p* < 0.001).

## 3. Results

From a psychometric standpoint, all variables assessed in the study showed satisfactory Cronbach’s alphas ranging between 0.73 and 0.90. Moreover, discriminant validity was confirmed through the Fornell–Larcker criterion and the HTMT, as showed in Table 1 and Table 2.

Table 1 and Table 2 highlight how discriminant validity was confirmed. In particular, following Henseler et al. [131], and Ab Hamid, Sami, and Sidek [137], in order to prevent multicollinearity of latent variables and to verify that they were measuring different constructs without overlapping each other, the Fornell–Larcker criterion and Heterotrait–monotrait (HTMT) ratio of correlation were performed. More specifically, by comparing the square root of the Average Variance Extracted and the correlation of latent variables the Fornell–Larcker criterion (Table 1) suggested that the first should be higher than the second one [138]. Furthermore, the discriminant validity of the latent dimensions of the study was corroborated by HTMT method (Table 2). As suggested in Henseler et al. [131], value of HTMT close to 1 denote the lack of discriminant validity—Kline [139] recommend a threshold of 0.85. Outputs of Table 1 and Table 2 point out that, according to these evaluation’s criteria, discriminant validity was confirmed. 

As for correlations (Table 3), data were consistent. More specifically, organizational communication showed a significant positive correlation with self-efficacy (*r* = 0.22), and a negative correlation with psycho-physical disorders (*r* = −0.17), and with technostress (*r* = −0.10). As for the technostress variable, beyond the correlation with organizational communication, it showed a positive and significant correlation with psycho-physical disorders (*r* = 0.35). 

The estimated Structural Equations Model showed satisfactory fit indices, which confirmed the goodness of the model fit: χ^2^(142) = 258.908, *p* < 0.00, CFI = 0.98, TLI = 0.97, RMSEA = 0.04; C.I. 95% (03; 05); SRMR = 0.03. Moreover, the Structural Equations Model showed parcels with significant loadings (*p* < 0.001). As for items loading in the model, even if organizational communication showed some low values, they can be considered acceptable [140]. 

Considering the wider research model (Figure 2), organizational communication was directly associated with all variables, in particular it was directly and negatively associated with technostress (*β* = −0.38), and with psycho-physical disorders (*β* = −0.18), and directly and positively associated with self-efficacy (*β* = 0.23), thus confirming, respectively, hypothesis 3a and hypotheses 1 and 3b. Technostress showed direct and positive and significant associations with psycho-physical disorders (*β* = 0.27), confirming hypothesis 2a, and self-efficacy showed a direct negative and significant association with psycho-physical disorders (*β* = −0.20), confirming hypothesis 2b.

The model explained the 22% of the variation in psycho-physical disorders, the 15% in technostress, and the 5% in self-efficacy. Even if this last value was weak, the fit indices of the Structural Equations Model can be considered excellent and the model was identified.

Moreover, in this study, technostress and self-efficacy confirmed their mediating role. The model, indeed, showed negative and significant and indirect associations between organizational communication and psycho-physical disorders through technostress (*β* = −0.11), and between organizational communication and psycho-physical disorders through self-efficacy (*β* = −0.05), confirming hypotheses 5a and 5b. Table 4 showed these statistically significant indirect effects, obtained with the bootstrapping procedure.

## 4. Discussion

This study focused on organizational communication to understand its role in relation to a new source of stress, the technostress, to an important personal resource, such as self-efficacy, and the possible relation with a negative outcome such as the workers’ psychological disorders. 

Considering that the data analyzed in the study were collected during the lockdown imposed by the Italian Government to deal with the emergency of the COVID-19 pandemic, findings contributed to extend both the literature and knowledge on this area of study which is rapidly growing. Therefore, to understand the psycho-social dynamics linked to the mandatory, unexpected and abrupt use of technology and to investigate the role of the elements that can buffer its impact and the consequences on individuals’ health, it could be crucial to design specific organizational interventions in the light with effective performance and workers’ well-being. 

As for the present study, all the hypotheses were confirmed. Specifically, according to hypothesis 1 a negative association between organizational communication and psycho-physical disorders was found. These findings allowed to capture the crucial role of a clear and punctual communication, especially during this period of crisis brought about by the pandemic. Yet, in times of troubles and change like the present one, many evidences confirm that communication might strongly influence employees’ commitment and their positive approach in coping with the need to re-organize work practices [19,23]. Moreover, within this frame, a crucial role could be played by the authority (e.g., managers, supervisors, employers) who can encourage employees to have trust and to be resilient through a clear and supportive communication aimed at managing emotionally demanding situations like the one experienced during this pandemic [141]. Moreover, in line with the Job Demands–Resources model, organizational support could be functional to improve engagement and to buffer discomfort outcomes. Organizational support can be conveyed and reinforced also through a positive communication, that especially in times of crisis [142], could reassure employees about their performance and encourage them to make ever better [95]. 

The crucial role of organizational communication also appeared in confirming hypotheses 2 (both 2a and 2b) and 3 (both 3a and 3b) explained as follows. In relation to hypotheses focusing on technostress, it is important to consider that technostress, which is a source of stress for employees who needed to be always on in a highly demanding and multitasking situation, was related to psychological discomfort, anxiety, and physical disorders, a relation confirmed in this study in hypothesis 2a. As organizational communication is found to be negatively associated with technostress, confirming hypothesis 3a, communication was proved to be a key well-being antecedent: it has the potential to support employees in efficiently coping with work stressors [82] and with organizational change [83]. Evidently, managing new working practices and processes also through the adoption of e-working modalities from home, might represent a challenge for technostress. However, this study contributed to highlight that organizational communication could be a positive resource in decreasing the stress deriving from technology and psycho-physical disorders. As technostress is linked to the risk of physical disorders [53], psychological disorders and also burnout [50], it is important to prevent possible negative outcomes. It is important, indeed, being aware of the possibility that these symptoms would develop and settle in the long period, and that they would have detrimental consequences for the individual on a health level, and for the organization on a productivity level. Studies on the psychological exhaustion, indeed, describe it as a long-term consequence after strain and high exposure to job demands [143] that might have an influence on the low investment of energy and on the reduced performance behaviors at work [144]. As communication was proved to have a significant role in the management and prevention of stress, communication can be an opportunity because it can engage individuals in relationship based on trust and empathy [142]. In this case, this can be useful to prevent the damages caused by technostress, preventing psycho-physical discomfort and enhancing performance. This is also in line with suggestions related to a supportive and responsive communication that can cultivate positive emotional culture in organizations [145] with positive consequences in the long term for both individuals and organizations. 

In terms of positive consequences, this is in line with the results on self-efficacy which confirmed the related hypotheses in this study. In particular, a negative relation between self-efficacy and psycho-physical disorders found in this study, confirming hypothesis 2b. As underlined by studies, the presence of anxiety, depression, physical illness, and low performance, are related to a lack of self-efficacy [62] and of control over a working situation [63]. Self-efficacy should be considered as a positive antecedent of performance, but it is also a valid support in the prevention of discomfort, in line with the Job Demands–Resources Model and the possibility to identify self-efficacy as a personal resource. Personal resources are positive aspects of the self, linked to resilience and to the ability of individuals to control and manage their environment [59]. Personal resources have positive effects on psychological and physical well-being and help people to be able to deal with demanding situations, keeping them energetic, facilitating their engagement, and protecting them from psychological discomfort [59,146].

As hypothesis 3b, it is confirmed in this study. As already shown, communication can enhance self-efficacy also because it gives clear indications, offering therefore the perception of control over the situation. The possibility to positively manage organizational communication could be functional to enhance employees’ self-efficacy, and to reduce their anxiety and any other kind of psycho-physical disorders that in a period of crisis may arise [53], highly impacting on subjective well-being, and on performance and productivity as well. Communication, indeed, can be considered an organizational resource able to enhance personal resources in a sort of virtuous circle useful for the individual’s development [110,146]. It has to be underlined that resources, both job and personal, are linked to well-being [147], a concept in line with the Conservation of Resources theory [148], suggesting that people protect and maintain resources, creating new resources and thus resulting in positive outcomes and well-being. In this sense, it emerges again the key role of communication in acting as strong antecedent of this personal resource, and they mutually act on the decreasing of discomfort. This mutual action is also evident in the confirmation of hypothesis 4b, which stated that the association between organizational communication and psycho-physical disorders can be mediated, in a negative sense, by self-efficacy. Even if the impact is weak, it is significant and it is in line with studies suggesting the mediating role of personal resources between job resources and negative outcomes [146] and with the assumption of the Job Demands–Resources Model suggesting that resources stimulate personal development and growth [57]. This is an important point because it shows how the possibility to enhance a personal resource (in this case with communication as an antecedent of self-efficacy) can flow into positive outcome or buffer negative ones. This is functional to have the resources to deal with the situation and to prevent the possibility of stress [68,69]. In this sense, this relation could explain the virtuous circle between communication, and self-efficacy as a personal resource, in the reduction of discomfort. To conclude, considering that: first, communication can improve the sense of control, the development of skills [25,26,100,101], and is effective in coping with work stressors [82] in a well-being dynamic; second, self-efficacy is a personal resource with a positive impact on psychological and physical well-being [60,61], hypothesis 4b seems to confirm an intrinsic relationship between these variables and how they act in buffering discomfort outcomes.

Another important point of the indirect effects found in this study, is the action of communication in decreasing psycho-physical discomfort through technostress, which confirmed hypothesis 4a. The indirect effect is, indeed, weaker if compared to the direct effect from the variable of communication to the variable of psycho-physical disorders. The interesting element is that if technostress can strongly increase psycho-physical disorders, when there is the communication as antecedent, this relation lapses and communication has the power to stop the action of technostress on the employees’ discomfort. This is in line with studies suggesting the role of communication facing with demanding situations, but it represents also an important contribution to literature on the role of communication in the reduction of negative effect of technostress. In the light with these results, it is possible to frame organizational communication as a protective element for the employees’ well-being.

### Limitations

The first limitation of this study was the use of a cross-sectional design that did not allow to define causality relationships between variables. To better investigate the role of communication with respect to different tasks and to the level of the related technostress, it could be helpful to conduct a longitudinal study, in particular to understand if constructs related to well-being could depend on the particular experience of job or of the situation [149]. Longitudinal analyses can be helpful in the understanding of the relation between communication and psycho-physical disorders, also observing stress fluctuations on a daily basis. Finally, future studies could address the role of communication in relation to other personal resources that can be helpful in the reduction of discomfort, helping both the individual at a health level, and the organization, at a productivity level. 

Another limitation of the study was represented by the risk of common method-bias [150] due to the use of only self-reported data. Future studies should consider also objective data, such as days of permission for discomfort reasons or virtual absenteeism (in terms of work permission not linked to health that would indicate a form of disengagement), in order to better link self-report data and organizational scenario at a human resources level. 

Moreover, related to the sample, this study cannot generalize conclusions since it used only Italian workers. Future studies should replicate it to understand the possibility that other countries have benefited from remote work, by reducing stress or by increasing the economic development taking advantage of the ability to manage work processes from private home.

Despite all the limitations described above, this was one of the first study considering the role of communication in its interaction with the technology dynamics in times of COVID-19 pandemic emergency, allowing further reflections about potential key protective factors. Considering the current scenario of the labor market, featured by the presence of (old and young) workers coping with new digital working modalities and therefore with new related learning demands, this study could be considered as a first contribution to the understanding of the new social, organizational and personal demands, and resources that have emerged following to the emergency. However, to better capture the complexity and heterogeneity of different cultures, professions and organizations (e.g., with specific reference to tasks, skills, organizational dimensions, shifts, etc.) in the use of communication and in facing this new working scenario, future studies should consider multi-group analysis to suggest focused practical implications. 

Moreover, in order to better understand the active role of employees and to develop also positive and protective practices, future studies should consider also the role of the individual self-efficacy related to the use of technology, which might impact on technostress [36].

## 5. Conclusions

This study contributed to literature since it not only confirmed the importance of communication in organizational processes and its role in the management of stress, but also its crucial role in decreasing the particular type of stress coming from the use of technology in the specific period of remote working due to the COVID-19 pandemic emergency. This situation was unexpected and, in a very short time, organizations and workers had to reorganize many home-based work processes, they had to learn how to use new technologies and to re-arrange long-distance relationships. According to the findings in this study, the role of communication, in this reorganization, is very important because it can reduce technostress and psycho-physical disorders highlighting the importance of clear information and engaging situation. Moreover, this study highlights the key role of communication in enhancing self-efficacy, opening the possibility to focus on communication as empowering the personal resources, which are recognized by literature as very important for the individual well-being. Relations in this study might suggest the role of communication as a protective factor, in particular in this period of sanitary emergency, against psychological and physical disorders, but also to protect workers by buffering the effect of technology on the perception of stress. In this sense, this study represents also an important contribution to the literature on the role of communication in the reduction of negative effect of technostress: in this period technology is more and more pervasive and to understand the role of communication, as a protective and powerful instrument to safeguard health and prevent discomfort, is very precious for organizations. 

For this reason, organizations might apply specific communication strategies, friendly and usable, to face with the possibility of informal communication, and in order to act in this preventive vision. Moreover, organizational communication can help employees to understand the meaning of the stressors [100] and this would be functional to make them aware of the risk of technology and of being always connected. Organizations are exposed to a growing use of technology and must be aware of the advantages but also of the associated risks. With this study it was possible to underline the potential of communication and job design should consider this aspect in order to build positive and healthy organizational culture. Employees would be helped in the psychological and physical disorders management and would be more productive, with positive outcomes for organizations, since technostress has been found to be related to a lower performance [52]. This would also be conveyed through specific training for employees and managers that enables people to know the risks of technology and how to protect themselves. This would involve also specific strategy linked to the communication through technology. 

In the light with the potential of an effective communication system in organization and with the possibility to get the need information or important information on time [123] and, considering the continuous changes that technology and the development of work processes are facing, setting a punctual communication through friendly interface is not sufficient. To ensure the effectiveness of the organizational communication, it is necessary to constantly monitor the internal satisfaction of the communication by employees, as well as the ease of retrieval of information by those who are at a distance, working remotely, implementing specific categories, and categorizations of the information. This implies an accurate survey of information needs, as well as a precise structuring of communication processes both at a structural and relational level. This would be functional to avoid even the information overload that can come from a total digitization of the work [151].

## Figures and Tables

**Figure 1 ijerph-18-03933-f001:**
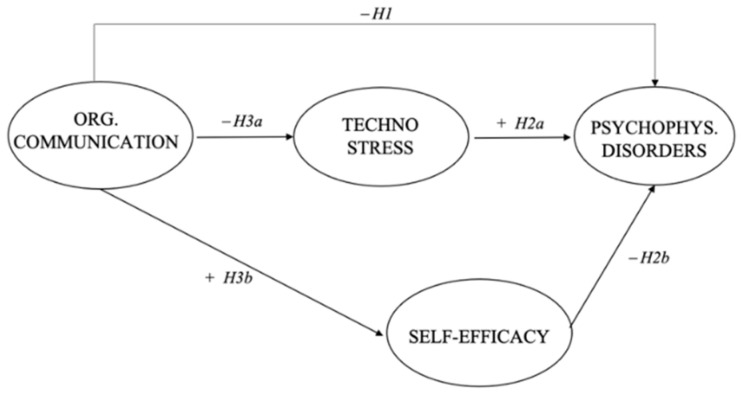
The hypothesized theoretical model.

**Figure 2 ijerph-18-03933-f002:**
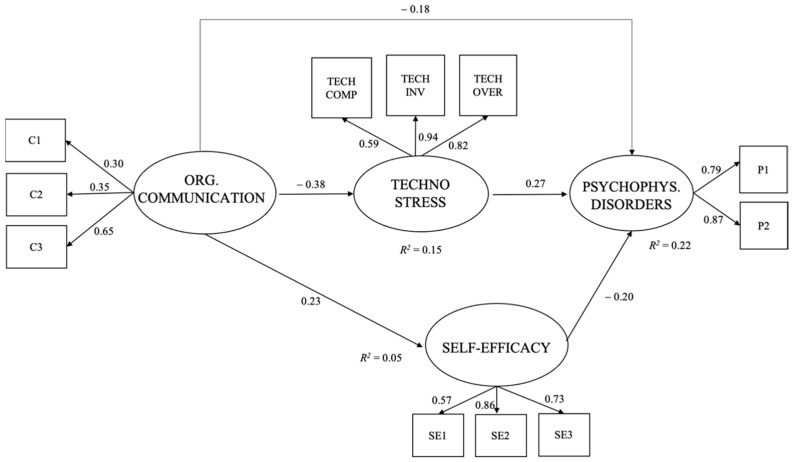
Results of the structural equations model. Note. TECH COMP = parcel of techno-complexity; TECH INV = parcel of techno-invasion; TECH OVER = parcel of techno-overload; C1 = item 1 of the latent variable organizational communication; C2 = item 2 of the latent variable organizational communication; C3 = item 3 of the latent variable organizational communication; SE1 = item 1 of the latent variable self-efficacy; SE2 = item 2 of the latent variable self-efficacy; SE3 = item 3 of the latent variable self-efficacy; P1 = parcel 1 of the latent variable psycho-physical disorders; P2 = parcel 2 of the latent variable psycho-physical disorders.

**Table 1 ijerph-18-03933-t001:** Fornell–Larcker criterion assessing of discriminant validity. Square root of the AVE on the diagonal.

	1	2	3	4
1. ORG. COMMUNICATION	*0.70*			
2. TECHNOSTRESS	−0.20	*0.71*		
3. SELF-EFFICACY	0.20	−0.11	*0.82*	
4. PSYCHO-PHYSICAL DISORDERS	−0.21	0.31	−0.22	*0.62*

**Table 2 ijerph-18-03933-t002:** Heterotrait–Monotrait (HTMT) ratio of correlation.

	1	2	3
1. ORG. COMMUNICATION	*-*		
2. TECHNOSTRESS	0.22	*-*	
3. SELF-EFFICACY	0.33	0.17	*-*
4. PSYCHO-PHYSICAL DISORDERS	0.21	0.33	0.27

**Table 3 ijerph-18-03933-t003:** Means, Standard Deviations, and Correlations (Pearson’s r).

	M	SD	1	2	3	4
1. ORG. COMMUNICATION	3.42	0.77	(0.73)			
2. TECHNOSTRESS	2.30	0.88	−0.10 *	(0.90)		
3. SELF-EFFICACY	4.13	0.77	0.22 **	−0.05	(0.76)	
4. PSYCHO-PHYSICAL DISORDERS	1.77	0.65	−0.17 **	0.35 **	−0.22 **	(0.84)

Note. ** *p* < 0.01 level; * *p* < 0.05 level. Cronbach’s alphas are on the diagonal (between brackets).

**Table 4 ijerph-18-03933-t004:** Indirect effects of the estimated SEM using bootstrapping (2000 replications).

Indirect Effects	Standardized Indirect Effects—Bootstrapping Procedure			
	Est.	s.e.	*p*	CI 95%
Organizational communication → Technostress → Psycho-physical disorders	−0.11	0.02	0.01	(−2.37, −0.21)
Organizational communication → Self-efficacy → Psycho-physical disorders	−0.05	0.03	0.00	(−5.419, −0.552)

## Data Availability

The data presented in this study are available on request from the corresponding author. The data are not publicly available in compliance with the protection of personal data and art. 9 of the Deontological Code of Italian Psychologists.

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
