# Peer review of "Does the End Justify the Means? The Role of Organizational Communication among Work-from-Home Employees during the COVID-19 Pandemic"

_ijerph, 2021, doi:10.3390/ijerph18083933_

Round 1

Reviewer 1 Report

Congratulations on your work. It is well-conceived, clear, and well-analyzed.

  1. Unless the authors assure that the world will never be like it was before COVID, I suggest that the authors remove or modify the word "irreversibly" (line 38)
  2. Please, correct some minor typing errors (e.g., modelsof, line 82; namelymessages, line 87; workaholism, line 138; hade, line 321; 2.000, line 384; objectve, line 557; recongized, line 587; organizazional, line 616; wolud, line 622)
  3. It is widely known the meaning of the acronym ICT (line 120), but you must expand it the first time you use it.
  4. The authors have written, "during the lockdown imposed 
    by the Italian Government" (line 325). I suggest the authors specify the duration of this lockdown (December 2020? January 2021?)
  5. I propose the authors eliminate "the two variables of" (line 437) to make reading clearer.
  6. I propose the authors introduce a new limitation in their study, as they have only used Italian workers in the sample. Although the results obtained are applicable in other countries, there may be exceptions in the case of very isolated regions of the world that have seen remote work as an opportunity for economic development, or in highly populated cities, where the resulting psychological and health problems of stress have been reduced by remote work.

Author Response

Dear Reviewer,

thank you for your suggestions.

Emanuela with the Authors

Reviewer 2 Report

Thank you for your interesting study about the role of organizational communication regarding psycho-physical disorders. I think the results are interesting and also include some sophisticated analyses, such as SEM and calculation of indirect effects. However, I think that the manuscript needs revising in the introduction and discussion part. Also, I have difficulties understanding the operationalization of organizational communication (see my comment number 5). All of my comments are listed below.

1) Before hypothesis 1 is stated, literature about the connection between organizational communication and self-efficacy has to be included. Self-efficacy is described in detail on pages 5 and 6, but this comes very late and should be stated before hypothesis 1 is introduced.

2) What is the difference between technostress and technostress creators? For example, in Hypothesis 2, the term technostress is used, while in the abstract, the authors refer to technostress creators. One term should be used in the manuscript to make the text more consistent. In the discussion, there is also a mix of both terms.

3) Please include an explanation (or a theory) why organizational communication is negatively related to technostress. Technostress is the stress that is experienced while working with technology. Which communication processes could lead to a reduction of technostress?

4) Hypothesis 5 is unclear to me. How exactly should the indirect effects work? I suppose that technostress strengthens the negative relationship between organizational communication and psycho-physical disorders. As for self-efficacy, it is ambiguous. Should self-efficacy weaken the negative relationship between organizational communication and psycho-physical disorders? Or should it strengthen the relationship as well? As both effects are described in one hypothesis, it is unclear in which direction the effects should be expected. It was also not evident to me when reading the discussion. The sentence “organizational communication is negatively associated with psycho-physical disorders through self-efficacy” does not explicitly state what exact role self-efficacy plays in the relationship between organizational communication and disorders. In comparison, the effect of technostress was well explained in the discussion..

5) Reading the introduction, I assumed that with the term “organizational communication” a more exchange-related approach towards communication is meant. That means an exchange of information between people within the company, perhaps also in a participatory approach, where employees can voice their interests. You emphasize this also in your text, for example on page page 3 line 107, or on page 5, line 198 (where you describe listening as an important part of communication). I agree with all of that. However, you used a scale of the COPSOQ to operationalize organizational communication, which is described as “predictability” in the COPSOQ. This scale measures if employees receive all important information for their work. To me, this means rather “information flow” than “organizational communication”. I am very curious to hear the authors' response.

6) In my view, the beginning of the discussion gives a much better insight of what is meant by organizational communication and how the concept should be framed. I think the introduction should describe the concept with similar clarity.

7) I would not say that heterogenous data are limited data. The fact that you’ve gathered data from different organizations and thus building a heterogenous sample is indeed a strength in your study, while a homogenous sample from one company would only allow very limited conclusions.

8) I would move most of the text in the conclusion (from line 567 and forward) to the section “practical implications”.

Other:

There a several typos and grammatical errors in the manuscript, for example:

Page 2, line 82: the organization is considered as a complex modelsof communication habits, (should mean model of)

Page 2, line 87: namelymessages exchanged between people who hold a different position within the organization between people who occupy the same position (“namely messages” and “organization or between people”)

Page 6, line 251: exposure to stressful factors has no negative consequences, until the person maintains high control (the word “until” is not the right term in this context, perhaps you mean “if”)

Page 6, line 286: “who were alone and already psychologically felt” (I think there is a word missing at the end).

Page 7, line 294: “that many participants reported having, during the period of  lockdown, headaches, indigestion and sleep irregularities.” (move “during the period of lockdown” to the beginning or the end of the sentence)

Page 13, line 607: “since has technostress has be found to be related” … “since technostress has been found to be related”

Please revise following sentences:

Page 5, line 212: how a forceful relationship between the perceived support and the communication style à what do you mean with “forceful relationship”?

Page 5, lines 213ff: Please rewrite the sentence where you explain the organizational support theory. The sentence is very complicated and hard to understand.

Page 5, lines 223ff: what do you mean with the term “brand” in this context?

Page 11 / Page 12, line 502ff: This sentence is also very complicated, for example “have a significant role in stress (and its outcome) management”… or “relationship with trust in and empathy”, should be revised

Page 13, line 556: “Future studies should consider also objectve data, such as day of permission for discomfort reasons or virtual absenteeism”. I suppose that days of permission for discomfort reasons are absenteeism rates, but what is meant with virtual absenteeism?

Author Response

(The authors gave the same response as above.)

Reviewer 3 Report

I want to thank the authors for this interesting submission on the role of organizational communication in individual well-being. While I like the general idea and how relevant it is to our current situation, I am doubtful whether the presented study is ready for publication due to several issues.

First, there are several major issues with how the study is conceptualized and how the manuscript is structured. An initial concern I had was that the way that the authors have argued for the different facets of organizational communication (e.g., formal vs informal, digital vs face-to-face) would not be reflected in how the construct is then actually treated in the study. Organizational communication is the main construct of this study, but is then only operationalized with a three-item scale, which in turn only shows weak loadings (Figure 2). As the main focus of the study, this construct has to be treated more seriously, which also includes an in-depth discussion of its facets and how they can be integrated into a higher-order construct.

Second, the way that theories are inserted as well as additional constructs is overall sloppy. For example, the Job Demands / Job Resources Model suddenly appears on p. 4, but without leading to further insight into how the relationship mentioned in H2 would form. Also, communication style or perceived organizational support are at some points related to organizational communication without leading to any related hypothesis or any related measurements. I would suggest that such interjections that do not offer additional, meaningful contributions, which actually affected the research model and the subsequent measurement model, should be removed.

Third, in line with how the Job Demands / Job Resources Model was suddenly introduced, the structure of the hypothesis building is not straight-forward. I would therefore suggest some changes regarding in which order they are presented starting with the effect of Org. Communication on psycho-physiology (H1), followed by the influence of two additional variables on psycho-physiology (H2a: Technostress; H2b: Self-efficacy), followed by the effect of Org. Communication on these two variables (H3a: Technostress; H3b: Self-efficacy) and finally the proposition of potential mediating effects (H4a and b). In this context it should also be mentioned that the selection of the utilized variables is not clear. For example, why did the authors focus on general self-efficacy, but omitted more specific technology self-efficacy, which has also been shown to influence technostress (Ayyagari et al. 2011)?

Fourth, there are also some important issues that concern the applied methodology. For one, the title of the paper references “work-from-home employees”, but it is not mentioned whether this focus is also reflected in the sampling strategy. Next, as already mentioned before, operationalizing the main construct of the investigation with a three-item construct that clearly omits many facets of the construct is a major oversight that leads me to believe that this construct was originally not the main focus of this investigation (perhaps just a control variable) which has then been “elevated” to its current status. In any case though, this operationalization is not in line with the level of theorizing that centers on this construct. Then, the authors should explain why they used different scales instead of normalizing them (e.g., 5-point and 6-point and with different anchors) – also, the authors should explain whether the scale for Organizational Communication (never to always) is actually comparable to the scale used for the other constructs (strongly disagree to strongly agree) as it is originally not conceptualized as a Likert scale. Next, in addition to reliability, also discriminant validity for the constructs should be demonstrated, for example using the Fornell-Larcker criterion (Fornell and Larcker 1981) and the HTMT (Henseler et al. 2015). Further, it is not explained how the authors interpret and handle and low loadings of the some of the items (e.g., .30 and .35 in the case of Organizational Communication) and it should also be explained how it is possible that the R² for Self-Efficacy can be .53 it is only exogenous variable has a correlation coefficient (R) with it of .22 (the R² should then be around .048).

Fifth, related to the physiological outcomes of technostress some previous has been published that they authors seem to have neglected fully (e.g., Galluch et al. 2015; Riedl 2013; Tams et al. 2014).

Sixth, the manuscript needs some serious proofreading. I usually outline the main passages throughout a reviewed paper that need some editing but in this case I gave up after the initial pages, with some examples including, but far not limited to (potential remedies in brackets):

  1. 2 / 53: “…by convey[ing] motivate[ing] messages in order to gain productive performance and a confident attitude.”
  2. 2 / 55: “…have [the] purpose to strengthen…”
  3. 2 / 70: “This study significantly contributes to the reading of such difficult period in which communication appears more and more important in organization, but the associated technology is being considering as a source of stress that can be responsible of the arising of new form of negative psychological and physical consequences.”

Finally, and just as a minor recommendation, the numbers of relevant hypotheses should be included in Figure 1.

Overall, I hope that the authors can profit from my comments and wish them all the best for their future research.

References

Ayyagari, R., Grover, V., and Purvis, R. 2011. “Technostress: Technological Antecedents and Implications,” MIS Quarterly (35:4), pp. 831–858.

Fornell, C., and Larcker, D. F. 1981. “Evaluating Structural Equation Models with Unobservable Variables and Measurement Error,” Journal of Marketing Research (18:1), p. 39.

Galluch, P., Grover, V., and Thatcher, J. B. 2015. “Interrupting the Workplace: Examining Stressors in an Information Technology Context,” Journal of the Association for Information Systems (16:1), pp. 1–47.

Henseler, J., Ringle, C. M., and Sarstedt, M. 2015. “A new Criterion for Assessing Discriminant Validity in variance-based structural equation modeling,” Journal of the Academy of Marketing Science (43:1), pp. 115–135.

Riedl, R. 2013. “On the Biology of Technostress: Literature Review and Research Agenda,” DATA BASE for Advances in Information Systems (44:1), pp. 18–55.

Tams, S., Hill, K., Ortiz de Guinea, A., Thatcher, J., and Grover, V. 2014. “NeuroIS - Alternative or Complement to Existing Methods? Illustrating the Holistic Effects of Neuroscience and Self-Reported Data in the Context of Technostress Research,” Journal of the Association for Information Systems (15:10), pp. 723–753.

Author Response

(The authors gave the same response as above.)

Round 2

Reviewer 2 Report

Thank you for rewriting the introduction and adding now more information about “organizational commitment” and “self-efficacy”. Especially regarding organizational commitment, I think the manuscript includes much better the concept and meaning behind this construct.

I have some minor suggestions regarding the manuscript:

Lines 125-127 do not help in understanding how communication can decrease psychophysical disorders or technostress, as relationships with job satisfaction, performance and stress are stated. While stress is connected to psychophysical disorders, job satisfaction and performance are different work-related parameters. The sentence before (lines 122-124) however do explain the relationship much better.

Page 5, line 238-239. Why is the emphasis so strong on “work family conflict”? This variable is not part of your study. If you want to say that there are other factors influencing physical disorders, I suggest that this should be part of the discussion.

Page 7, Line 305-307

In line with previous studies, communication is related with well-being outcomes, it is also recognized to be a potential buffer for technostress, since supporting workers with information helps the anxiety and discomfort associated with technology.

I suppose it helps reducing the anxiety and discomfort. So include “reducing” or “lowering” or any similar word.

Page 7, Line 321-323

In this view, it has also to be considered the role of resources in organization in enhancing personal resources in a positive relationship of continuous growing and development

Please rewrite this sentence. It is very hard to follow the argumentation.

Author Response

Dear Reviewer,

thank you for your suggestions. 

Emanuela with Authors

Reviewer 3 Report

I want to congratulate the authors for this extensive revision, which certainly improved the quality of the manuscript. The majority of my comments have been addressed sufficiently and for those areas which would have required more extensive changes (e.g., survey measures used), the authors provided a meaningful explanation of their approach. The theoretical underpinning in particular is now much better. While I still see methodological weaknesses, I consider them acceptable at this point, as they are at least mentioned as limitations and offer potential for future research. The only point that I would like the authors to still address is another round of proofreading for those text passages that have been added (e.g., “Gallugh et al.” on line 203 shows the need for this step).

Author Response

(The authors gave the same response as above.)
